# Direct Cell Death Induced by CD20 Monoclonal Antibodies on B Cell Lymphoma Cells Revealed by New Protocols of Analysis

**DOI:** 10.3390/cancers15041109

**Published:** 2023-02-09

**Authors:** Michael Constantinides, Alexis Fayd’herbe De Maudave, Marie Potier-Cartereau, Mauricio Campos-Mora, Guillaume Cartron, Martin Villalba

**Affiliations:** 1IRMB, University of Montpellier, INSERM, CHRU de Montpellier, F-34090 Montpellier, France; 2Département d’Hématologie Clinique, CHRU de Montpellier, F-34090 Montpellier, France; 3Inserm UMR 1069, Nutrition Croissance Cancer, Faculté de Médecine, Université de Tours, F-37032 Tours, France; Réseau 3MC “Molécules Marines, Métabolisme et Cancer” and Réseau CASTOR “Cancers des Tissus Hormono-Dépendants” Cancéropôle Grand Ouest, F-34090 Montpellier, France; 4IRMB, University of Montpellier, INSERM, CNRS, CHRU de Montpellier, F-34090 Montpellier, France

**Keywords:** CD20 mAbs, direct killing, obinutuzumab, rituximab, B cell lymphoma, LLC

## Abstract

**Simple Summary:**

CD20 monoclonal antibodies exert a strong ability to quickly kill B cells by using different mechanisms, both direct and indirect, and assessment of killing is difficult. Cytometry emerged as an attractive alternative to classic assays using radioactivity; however, we show here that common cytometry protocols involving centrifugation can lead to missing an important part of the fast killing effect of drugs as mAbs. We show here that rituximab and obinutuzumab direct killing is faster and more important than believed, and that those mAbs differently impact calcium mobilization depending on the time of treatment. We propose here alternative methods to assess fast target cell killing in vitro, avoiding centrifugation or based on survival comparison with control cells or counting beads, both on B cell lines or B-CLL and NHL primary cells.

**Abstract:**

CD20 monoclonal antibodies (mAbs) eliminate B cells in several clinical contexts. At least two of these Abs, obinutuzumab (OBI) and rituximab (RTX), induce quick elimination of targets and put cancer patients at risk of tumor lysis syndrome (TLS) within 12–24 h of the first dose. The mechanisms of killing can require the recruiting of effector mechanisms from the patient’s immune system, but they can induce direct killing as well. This can be more rapid than recruiting cellular effectors and/or complement. We showed here that OBI and RTX induce quick (<1 h) and high (up to 60% for OBI) killing of two different B cell lines. This was unveiled by using two different techniques that circumvent cell centrifugation steps: a Muse^®^ Cell Analyzer-based approach and a direct examination of the cells’ physical properties by using forward scatter (FS) area and side scatter (SS) area by flow cytometry. These results excluded the presence of aggregates and were also confirmed by developing a normalized survival ratio based on the co-incubation of RTX- and OBI-sensitive cells with MOLM-13, an insensitive cell line. Finally, this normalized survival ratio protocol confirmed the RTX- and OBI-direct killing on primary tumor B cells from B cell chronic lymphocytic leukemia (B-CLL) and Non-Hodgkin’s lymphoma (NHL) patients. Moreover, we unveiled that direct killing is higher than previously expected and absent in patients’ samples at relapse. We also observed that these mAbs, prior to increasing intracellular calcium levels, decrease calcium entry, although manipulating calcium levels did not affect their cytotoxicity. Altogether, our results show that direct killing is a major mechanism to induce cell death by RTX and OBI mAbs.

## 1. Introduction

Therapeutic monoclonal antibodies (mAbs) have revolutionized cancer treatment, especially hematological malignancies, improving their outcomes. The first CD20 mAb approved by the FDA in 1997 was rituximab (RTX). Its approval for treatment of B-Non-Hodgkin Lymphoma (NHL) paved the way for development of mAbs in hematological malignancies and, now, RTX and other CD20 mAbs are routinely used in the clinic for B cell depletion [1]. This success has led the interest in their clinical mechanism of action. They induce cell death in four main ways, three of which rely on recruiting effector mechanisms from the patient’s own immune system. They are antibody-dependent cell-mediated cytotoxicity (ADCC), complement-dependent cytotoxicity (CDC), and antibody-dependent phagocytosis (ADP). The fourth is direct killing [1,2]. Due to the reliance of these Abs on the patient’s immune system to mediate antitumor effects, their exact in vivo mechanisms remain challenging to study. CD20 mAbs are classified depending on the target epitope, binding characteristics, and the mechanism leading to target cell killing. Type I CD20 mAbs such as RTX bind to two CD20 molecules in two different CD20 tetramers, and induce lipid raft rearrangement and CD20 clustering into them leading to caspase-dependent apoptosis [1,2]. However, how this triggers caspase-dependent apoptosis is unknown, although the involvement of several signal transduction pathways (Src family kinases, Akt, ERK1/2, NF-κB, and p38 MAPK) has been proposed [2,3]. Type II CD20 mAbs bind to intra-tetrameric CD20 molecules and cause cell death through actin reorganization, leading to lysosome-mediated cell death, independently of caspases and lipid raft formation [4]. RTX injection into the cerebrospinal fluid, which probably contains limited immune cells, induces a reduction in central nervous system lymphoma [5], and the therapeutic effect can be augmented by concomitant injection of serum as a source of complement [6]. This supports a role of direct toxicity in vivo in humans, but there are no more clinical evidence [1,2]. Obinutuzumab (OBI), a recombinant type II CD20 **and** immunoglobulin G1 (IgG1) Fc-optimized mAb, is superior to RTX in follicular lymphoma (FL) and B cell chronic lymphocytic leukemia (B-CLL), but with increased toxicity [7]. OBI has increased direct killing compared to RTX [1]. Cancer patients treated with OBI or RTX are at risk of tumor lysis syndrome (TLS) within 12–24 h of the first dose, suggesting a quick effect that could be related to direct killing. For example, RTX can clear peripheral blood B cells by 90% in 24 h [8]. In this article, we gain an insight on RTX- and OBI-induced direct cell death within a relatively short time frame and describe flow cytometry-based methods to determine this direct killing.

## 2. Materials and Methods

### 2.1. Ethical Statement

The use of human specimens for scientific purposes was approved by the French National Ethics Committee. All methods were carried out in accordance with the approved guidelines and regulations of this committee. Written informed consent was obtained from each patient or donor prior to surgery. Data and samples from patients were collected at the Clinical Hematology Department of the “CHRU de Montpellier”, France, after patient’s written consent and following French regulations. Patients were enrolled in the HEMODIAG_2020 (ID-RCB: 2011-A00924-37) clinical program approved by the “Comité de Protection des Personnes Sud Méditerrannée I” with the reference 1324. Samples were collected at diagnosis and, in some cases, after relapse and kept by the “CHRU de Montpellier”.

### 2.2. Processing of Patient Samples

Patient’s samples were processed <24 h after blood collection. Peripheral blood was diluted with an equal volume of phosphate-buffered saline (PBS) and PBMCs (peripheral blood mononuclear cells) were obtained using Histopaque^®^ 1.077 (SIGMA). Samples were centrifugated for 30 min at 400 g and PBMCs were harvested, washed twice using PBS, and counted. Patient samples were frozen at −80 °C using a 1 °C/min temperature decrease rate, in 90% FBS + 10% DMSO. Samples were stored in liquid nitrogen until further use. Patients’ PBMCs were thawed and DMSO-washed with PBS. PBMCs were in culture for 4–8 h in complete culture media at 37 °C, 5% CO_2_ prior to the experiment.

### 2.3. Cell Lines and Cell Culture

Raji and Daudi (B cell lines derived from Burkitt lymphomas), and K-562 (derived from chronic myelogenous leukemia) were obtained from ATCC. MOLM-13 (derived from acute monocytic leukemia (AML-M5a)) were obtained from Dr. J.E. Sarry at Centre de Recherches en Cancérologie de Toulouse, UMR1037, Inserm, Université de Toulouse 3, France. All cells were cultured in complete culture media, consisting of RPMI 1640 Glutamax™ medium supplemented with 10% heat-inactivated FBS (both purchased from Gibco), at 37 °C in a humidified 5% CO_2_ atmosphere incubator in a BSL-2 facility room, without any antibiotics. All cell lines were periodically tested for potential mycoplasma contamination using a commercial kit (Mycoalert™ kit; Lonza, Basel, Switzerland). Prior to any medium change, cells were counted using trypan blue exclusion or a Muse^®^ cell counter device (Millipore, Burlington, MA, USA). 

### 2.4. Measurement of Direct Target Cells Killing with the Muse Device

The Muse^®^ Cell Analyzer uses miniaturized fluorescence detection and microcapillary cytometry to deliver single-cell analysis, which is highly quantitative. Here, we used the Muse^®^ Count & Viability Assay kit to unveil cell death following the manufacturer’s protocol, which does not require centrifugation. Cells were counted using the Muse cell counter device (Millipore). A total of 200,000 target cells were placed in 96-well U bottom plates. Plates were centrifugated (300 g, 5 min) and supernatant was removed. We added 100 µL of complete culture media (see Section 2.3) with or without RTX/OBI (10 µL/mL). Plates were placed in an incubator (37 °C, 5% CO_2_) for different incubation times. At the end of the chosen incubation time, cells were resuspended, 20 µL was sampled, and 180 µL of Muse^®^ count and viability buffer was added 2 min prior to measurement using a Muse^®^ cell counter device (Millipore). 

### 2.5. Measurement of Cell Death, without Staining/Centrifugation, Using Gallios 3L

Two hundred thousand target cells were placed in 96-well U bottom plates, which were centrifugated (300 g, 5 min) to remove the supernatant. Next, we added 100 µL of complete culture media (see Section 2.3) with or without RTX/OBI (10 µL/mL). Plates were placed in an incubator (37 °C, 5% CO_2_) for different incubation times and directly transferred to FAC tubes for analysis in a Gallios 3L as described in Appendix A.

### 2.6. Measurement of Cell Death by Staining and Cytometry Assay

The Viobility™ 405/452 and the 7-Aminoactinomycine D (7AAD) were utilized according to manufacturer’s instructions. After incubation, cells were washed in PBS and centrifugated for 5 min at 300 g and stained in 20 µL of staining medium for 20 min at RT in the dark and then resuspend in PBS and wash. For patients’ samples, we used Viobility™ 405/520 instead of viability™ 405/452, following the same protocol. 7AAD, Viobility™ 405/452, and cytotrackers CellTrace™ Far Red and CFSE (CTFR and CTG) fluorescence were, respectively, observed on FL4, FL9, FL6, and FL1 channels (Gallios 3L, 12 parameters, Beckman Coulter, Brea, CA, USA). Viobility™ 405/520 was observed on channel BV510 on FACSymphony A3 (BD Biosciences, San Jose, CA, USA). Appendix A gives details about this analysis.

### 2.7. Measurement of Cell Death by Using an Unsensitive, Control, Cell Line

Target cells and MOLM-13 cells were stained at 1.4 × 10^6^/mL in RPMI1640, respectively, with CellTrace™ FarRed (Invitrogen, Waltham, MA, USA, ref: C34564) diluted to 1:2500 or CellTrace™ CFSE (Invitrogen, ref: C34554) diluted to 1:5000 after washing with PBS. After 20 min of labeling, cells were washed 3 times and placed in complete culture media (see Section 2.3). Next, 200,000 target cells and 200,000 MOLM-13 cells were placed in 96-well U bottom plates. Plates were centrifugated (300 g, 5 min) and 100 µL of complete culture media (see Section 2.3) with or without RTX/OBI (10 µg/mL) was added. Plates were placed in an incubator (37 °C, 5%CO_2_) for different incubation times. Appendix A explains how the normalized survival ratio of the target cells was quantified.

### 2.8. CD20 Quantification

CD20 quantification was performed using a Quantibrite approach using PE-conjugated beads (BD Quantibrite™ Beads, ref: 340495; batch: 40966), by adding 10 µL in 100 µL of CD20 PE-conjugated mouse IgG1K antibody (clone B27, BD biosciences, ref: 347201) on 200,000 cells, incubated at 4 °C for 20 min prior to two washing steps using PBS + 2% FBS. We used Her2 mouse IgG1 PE-conjugated antibody (clone NEU 24.7, ref: 340552) as the control isotype. We used Filter 2 (575/30 nm) on a Gallios 3L cytometer (Beckman Coulter) using the same settings for all Geometrical Means. Downstream analyses were performed according to the provider’s instructions.

### 2.9. Tumor B Cell Death Assessment by Using Beads

After thawing and the short recovery time in complete culture media, patients’ PBMCs were counted and viability assessed using the Muse device. Next, 200,000 cells were seeded in 96-well U bottom plates, centrifugated (300 g, 5 min), and resuspended in 100 µL of complete culture media (see Section 2.3) with or without RTX/OBI (10 µL/mL). After incubation, cells were centrifugated (300 g, 5 min), washed, and stained for cell dead assessment using Viobility 405/520 as previously described. After staining, cells were washed and resuspended in 20 µL/well using the following antibody cocktail: 1:200 hCD19-FITC (Miltenyi, Bergisch Gladbach, Germany, clone REA 675, ref: 130-113-645), 1:50 CD20-PE (BD biosciences, ref: 347201), 1:200 hCD10-BUV496 (BD biosciences, clone MEM-78, ref: 750190), 1/200° hCD5-BUV737 (BD biosciences, clone: UCHT2, ref: 612842), 1:200 hCD23-BV711 (BD biosciences, clone: EBUCS-5, ref: 743430), 1:200 hCD79a-PE-vio770 (Miltenyi, clone: HM47, ref: 130-104-226), 1:200 hKappa Light chain-BUV395 (BD biosciences, clone G-20-193 ref: 743168), and 1:200 hLambda Light chain-BUV805 (BD biosciences, clone: JDC-12, ref: 748804), in 90% (PBS 2% FBS) + 10% Fc Block (Miltenyi, ref: 130-059-901). Cells were incubated for 20 min, washed twice in PBS +2% FBS (centrifugation 500 g, 5 min each time), and resuspended in 180 µL of PBS 2% FBS plus 20 µL of precision counting beads (Biolegend, San Diego, CA, USA, ref: 424902). Samples were analyzed using a FACSymphony A3 cytometer. The spillover matrix, tumor B cells selection, and viability assessment were conducted according to Appendix A.

### 2.10. Constitutive Ca^2+^ Entry Measurements

Cells were seeded at 600,000 cells in tissue culture dishes 48 h before the experiment and maintained in complete culture media (see Section 2.3). Cells were loaded in Petri dishes with ratiometric dye Fura2-AM (5 µM) with or without mAbs at the concentration of 10 µg/mL for 45 min at 37 °C. Then, cells were washed with RPMI1640 Glutamax media without serum and centrifugated (×500 g for 2 min). Immediately after centrifugation, cells were re-suspended in 2 mL of PBS Ca^2+^-free solution containing (mM): NaCl, 140; MgCl2, 1; KCl, 4; EGTA, 1; D-glucose, 11.1; HEPES, 10, adjusted to pH 7.4 with NaOH. After 8 s, 5 mM of calcium was applied. The amplitude of constitutive calcium entry was measured and analyses were performed using the SoftMax Pro Software (Molecular Devices, San José, CA, USA). For the second protocol, cells were suspended in PBS without Ca^2+^, Mabs were added at the beginning of the measurement (acute treatment), and after 8 s, 5 mM Ca^2+^ was applied. Intracellular Ca^2+^ variations were evaluated using the fluorescence emission measured at 510 nm with an excitation light at 340 and 380 nm (Hitachi, FL-2710).

### 2.11. Calcium Deprivation and Calcium Ionophore Impact on Direct Killing by CD20 mAbs

Cells were seeded at 300,000 cells per well in 96-well U bottom plates, centrifugated (300 g, 5 min), and resuspended in 100 µL of complete culture media (see Section 2.3), which was freshly supplemented with different concentrations of ethylene glycol-bis(β-aminoethyl ether)-N,N,N′,N′-tetraacetic acid (EGTA; range 0 to 10 nM), with or without RTX/OBI (10 µL/mL) in the presence of ionomycin (1 µg/mL) and phorbol 12-myristate 13-acetate (PMA) (50 ng/mL). After resuspension, cells were incubated for 1 h and viability was quantified using Muse^®^ count and viability buffer as described in Section 2.4.

### 2.12. Cytometry Experiments

Acquisition was performed using Kaluza acquisition software V1.3 or BD Facs Diva, depending on the cytometer used for each experiment. Analyses were performed using Kaluza analysis V2.1 software analysis.

### 2.13. Statistical Analysis

Statistics were performed using Prism V7.04 software. Each experiment was performed in duplicate. Mean, standard deviation, and number of replicates for each condition were extracted and placed in a matched manner to compare the impact of treatment. The statistical tests were performed as detailed in each figure legend.

## 3. Results

### 3.1. RTX and OBI Induced Cell Death on CD20+ Cells One Hour after Treatment: A Muse^®^ Cell Analyzer Protocol

We used the kit Muse^TM^ Count & Viability Assay, which does not require centrifugation, to study cell death after CD20 mAb treatment. This was performed in culture media containing decomplemented serum to avoid CDC. We incubated 2 tumor B cell lines, i.e., Raji and Daudi, for up to 24 h with RTX or OBI. RTX induced cell death 1 h after incubation, and longer incubations, i.e., 2 and 4 h, increased cell death (Figure 1, left panels). We also observed this increase at 8 and 24 h post-treatment (Figure 1, right panels). OBI-mediated cell death, which was higher than RTX-induced, was already at its maximum 1 h after treatment (Figure 1). RTX and OBI did not affect viability of the cell lines K-562 and MOLM-13, which lack CD20 (Appendix A).

### 3.2. RTX and OBI Induced Cell Death on CD20+ Cells One Hour after Treatment: A FACs Protocol

This rapid killing was somehow unexpected. Hence, we treated again the cell lines and analyzed them by flow cytometry after staining with two commonly used viability dyes: 7-aminoactinomycine D (7-AAD), a fluorescent chemical compound with strong affinity for DNA in cells with damaged membrane integrity; Viobility dye 405–452, which reacts with the primary amino groups of proteins. We also observed a rapid cell killing, which was absent in K-562 cells (Figure 2). However, B cell cytotoxicity was lesser than when using the Muse^®^ Cell Analyzer, and there was an increase in OBI-induced killing after 1 h of treatment. 

### 3.3. RTX and OBI Induced Cell Death on CD20+ Cells One Hour after Treatment: A FACs Protocol without Centrifugation

We suspected that differences on direct toxicity measured by both protocols could be due to centrifugation steps that are required by FACs protocols using fluorescent dyes. It is worth mentioning that the centrifugation speed used for cell wash and recovery might not be enough to decant dead cells, which could have changed their cell structure and/or generated cell debris. We treated cells as described above and analyzed the samples without centrifugation, by using a gating protocol that simply eliminated the very small debris and directly analyzed cell death by changes in forward scatter (FS) area and side scatter (SS) area parameters (Appendix A). RTX and OBI induced a large decrease in the number of events in the living cells gate that correlated with an increase in the dead cells and debris gate (Figure 3). In addition, we observed a low percentage of cell aggregates using this protocol (Appendix A), and it could not account for the lack of target cell population that we have observed. This excluded that our observations could be due to CD20 mAbs-induced aggregation of CD20+ cells as it had been proposed before [9]. This strategy allowed for clearly determining the percentage of living cells after mAb treatment (Appendix A). We also evaluated this protocol in freshly thawed cells. Although the initial percentage of dead cells was much higher, we still observed a quick mAb-induced cell death (Appendix A). Hence, these results suggest that this protocol is advantageous to analyze frozen samples from patients (see below). In addition, by using this strategy, we confirmed that K-562 cells were insensitive to RTX and OBI(Figure 3 and Appendix A).

### 3.4. RTX and OBI Induced Cell Death on CD20+ Cells: A FACs Protocol with Control Cells

We next decided to evaluate CD20 mAb direct killing by using an alternative approach, based on the incubation of target cells with non-target cells to generate a normalized survival ratio. This protocol offers an internal, Abs-insensitive, control to CD20 mAb while allowing centrifugation and, hence, the use of standard staining with multiple antibodies to target cell antigens (Appendix A). If target cells, but not control cells, decrease their number/frequency on the assay, we can assume that the treatment specifically eliminates the target cell population. We used this protocol to look at target cell survival at 8 h. Daudi cells were labeled with CTFR, whereas resistant MOLM-13 cells were incubated with CTG. Both cells were mixed at a 1-to-1 ratio and treated with RTX or OBI. Viability was evaluated by 7AAD and Viobility dyes and standard FACs analyses. We analyzed a gate called “single events”, which included all the events except doublets and very small debris (Appendix A). The percentage of 7AAD or Viobility-positive events gives the “standard” percentage of cell death. This was low in non-treated Daudi cells (2.2%; Appendix A) and increased after RTX treatment (43.5%; Appendix A). Therefore, the increase due to treatment was 41.7%. In the resistant MOLM-13 cells control, mortality was 3.2% and increased to 4.8% after treatment, hence a 1.5% increase. These values were similar to those described in Figure 2.

We next analyzed a smaller FS/SS gate called “cells”, which included only single events with the characteristic FS/SS parameters of alive cells (Appendix A). As expected, the number of dead cells in this gate was remarkably low (0.44% and 0.14% for Daudi and MOLM-13 respectively; Appendix A). After RTX treatment, there was a significant reduction in CTFR+ cells in the “cells” gate (Appendix A). The percentage of alive MOLM-13 cells was basically unchanged 0.16% (Appendix A), while this percentage increased to 4.32% for Daudi cells.

We next calculated the ratio of alive cells between target and control cells (Appendix A). Of note, we subtracted the cell death of the total cell number. For non-treated cells, the survival ratio was 0.8 and, for RTX-treated cells, it was 0.16 (Appendix A). Hence, the normalized survival ratio (Appendix A) was 0.2 (Appendix A). This meant a specific killing of 80% due to treatment. This value was much closer to the values obtained on analysis using protocols that did not involve centrifugation, such as those shown in Figure 1 and Figure 3.

We observed that the number of dead cells was increased by RTX and OBI in both Raji and Daudi (Figure 4 and Appendix A). The percentage of dead cells reached a maximum of 25%. In contrast, when we investigated cell death by using our ratio protocol, we observed a minimum of 50% of target cells missing (Figure 4 and Appendix A). As a control, when we used the K-562 cells, we did not observe any changes in viability or ratio by CD20 mAb treatment (Figure 4 and Appendix A). Hence, this strategy gave the same outcome than previous approaches that avoided centrifugation. Analysis of the ratio could be used in complex/heterogeneous samples such as primary cells or protocols to investigate NK-cell-mediated ADCC.

### 3.5. Cells That Survive to CD20 mAbs Treatment Showed Low CD20 Expression

To investigate the nature of the cells that survive CD20 mAb treatment, we analyzed them by using BD Quantibrite™ Beads. We found that 3 days after treatment, cells that had survived expressed lower levels of CD20 than at the beginning of the treatment (Appendix A). This observation suggests that treatment could downregulate CD20 expression, or that it selects cell clones expressing lower surface levels of CD20. This is in accordance to previously described reports [10,11].

### 3.6. Primary Tumor B Cells Were Very Sensitive to Direct CD20 mAb Killing

We next evaluated if RTX and OBI induced direct cytotoxicity on primary B tumor cells. We created a panel to identify the tumor cells in patient’s samples (Appendix A) that we validated in an NHL sample (Appendix A). To evaluate rapid direct cell death, we used B-CLL samples that contain, on average, 95% of CD19+ cells, and most of them are tumor B cells (Appendix A). This allows for investigating the B-CLL sample as a “pure” population. First, we observed that RTX and mainly OBI decreased the proportion of singlets (Figure 5a), strongly suggesting that these mAbs induced aggregation in primary B-CLL samples as previously suggested [9]. Analyzing cell toxicity on these samples is challenging because after thawing, the proportion of basal cell death was relatively high (Figure 5b). We still observed a quick direct cell toxicity induced by both, RTX and OBI (Figure 5b), and by subtracting basal cell death after thawing, we observed that cell death also occurred quickly, i.e., 1 h, after treatment and longer incubations had low or not effect (Figure 5c).

To further investigate the effect of CD20 mAbs in primary tumor B cells, we took advantage of the protocol described in Appendix A. However, adding MOLM-13 cells would render the tumor sample more complex; hence, we used beads as an internal control of cell survival and developed a normalized survival ratio (Appendix A). This gave us values of cell death higher than those obtained with the classical approach of cell viability (Appendix A). Because staining and processing samples with our panel of antibodies and FACs analysis requires at least 2 h, we decided to investigate cell death 8 h after treatment. RTX induced on average 20% of cell death in samples from NHL and none in samples from B-CLL patients when the cell viability protocol was used (Figure 6a,b, left panels). With the same samples, the normalized survival ratio showed 40% and 30% of cell death (Figure 6a,b, right panels). OBI induced, on average, 45% of cell death in samples from NHL and none in samples from B-CLL patients when the cell viability protocol was used (Figure 6a,b, left panels). With the same samples, the normalized survival ratio showed 65% and 40% of cell death, respectively, for B-CLL and NHL (Figure 6a,b, right panels).

During collection of patient samples, three of our NHL patients experienced relapse after treatment with CD20 mAb-based therapy. Therefore, we decided to collect samples from those patients and use their samples to compare the cell sensibility to CD20 mAb treatment before and after relapse. Using the same approaches as in Figure 6, we observed that the samples were sensitive to direct killing at diagnosis, but insensitive after relapse (Figure 7). These differences where statistically different when we used the normalized survival ratio and were not while using the cell viability protocol.

However, the extent of mAb-induced direct killing, measured by viability assay or by normalized survival ratio, did not correlate with the level of the CD20 expression by primary cells (Appendix A).

### 3.7. RTX and OBI Affect Constitutive Calcium Entry

RTX and OBI increase store-operated calcium entry and this is involved in their mechanism of cytotoxicity [12]. We observed that these mAbs also increased constitutive calcium entry (CCE) 45 min post-treatment in CD20^+^ cell lines (Figure 8a). We observed cell killing in less than 1 h post-treatment. Hence, we analyzed cytosolic calcium variations just after treatment by measuring CCE. Both mAbs strongly decreased CCE in target cells (Figure 8b). To investigate if the variation in cytosolic calcium concentration could be linked to direct killing, we first treated target cells with RTX and OBI in the presence of the calcium ionophore ionomycin and of PMA (Appendix A). Ionomycin and/or PMA did not affect CD20 mAb killing. We next used the calcium chelating agent EGTA, which decreases extracellular calcium levels and can lead to a decrease in intracellular calcium levels [13]. EGTA did not affect mAb-induced killing (Appendix A). In summary, manipulating intracellular or extracellular calcium levels does not affect RTX and OBI-induced cytotoxicity in our experimental conditions.

## 4. Discussion

While assaying RTX- and OBI-induced cell death at timepoints routinely used by many researchers [14,15,16], including us, we realized that treated cultures had fewer cells than control, non-treated, cultures. We contemplated the possibility that centrifugation, a step that is normally required for washing cells before and after labeling with most dyes or Abs, could affect cell count, mainly related to the loss of cell debris. We used three different technical approaches to investigate this phenomenon: (i) the Muse^®^ Cell Analyzer; (ii) a regular cytometer, but avoiding centrifugation; iii) using ratios of survival between CD20-sensitive and non-sensitive cells. We unveiled that direct killing by RTX and OBI was quicker and more substantial than previously thought. The use of classical protocols (i.e., cell viability) failed to uncover the magnitude of such cell death. Whether our results are linked to the massive B cell death that occurs in patients within 12–24 h of the first RTX/OBI dose [17] is pending further research. However, it is tempting to speculate that such short and massive cell death is not only mediated by immune effectors, which, in most clinical situations, must be overwhelmed by tumor burden.

The mechanism of direct killing is different for both Abs [1,2]. RTX killing involves several signaling pathways including p38 MAPK, NF-kB, ERK1/2, and Akt and caspases [2,3]. OBI causes direct killing through actin reorganization, leading to lysosome-mediated cell death, independently of caspases and lipid raft formation [4]. Both processes are supposed to take a certain time to induce cell debris. However, the direct killing described here occurs in less than one hour. This suggests that the mechanisms described above should happen very quickly or not be implicated in our observations.

Assessment of direct killing by CD20 Abs is difficult to evaluate in several contexts. There is no standard technique for this assessment. Cytometry has been proposed in replacement of ^51^Cr methods to avoid radioactivity [18,19]. Previous works showed up to 40% and 60% of cell death 24 h after treatment with OBI and RTX, respectively, by using annexin V, propidium iodide (PI), and flow cytometric analysis [20]. These results have been challenged by the possible aggregation induced by type II CD20 antibodies [9]. In the present work, we have found a quick direct killing and, moreover, the absence of aggregation on cell lines, or low aggregation in primary samples. Therefore, we are confident that our results reflect CD20^+^ cell killing.

Some evidence supports a role for Fc receptors during RTX and OBI therapy, suggesting that ADCC is an important mechanism of action. However, despite investigations for over two decades, the pathways that contribute most to CD20 mAbs killing are yet to be fully elucidated. Our results could give new possibilities to explore this process in patients, for the most part, shortly after the first dose of treatment, when direct killing would be more relevant. The necessity of effector immune cells could be required in a second phase when surviving cells downregulate, or lowly express, CD20 molecules (our results and [10,11]). In this situation, an active killing could be essential for B cell elimination.

## 5. Conclusions

We have developed new protocols to evaluate the direct, and quick, killing of cytotoxic drugs. Using this methodology, we uncover that CD20 antibodies induce a much larger direct killing than previously described. Our finding could explain the rapid elimination of B cells in CD20 mAb-treated patients.

## Figures and Tables

**Figure 1 cancers-15-01109-f001:**
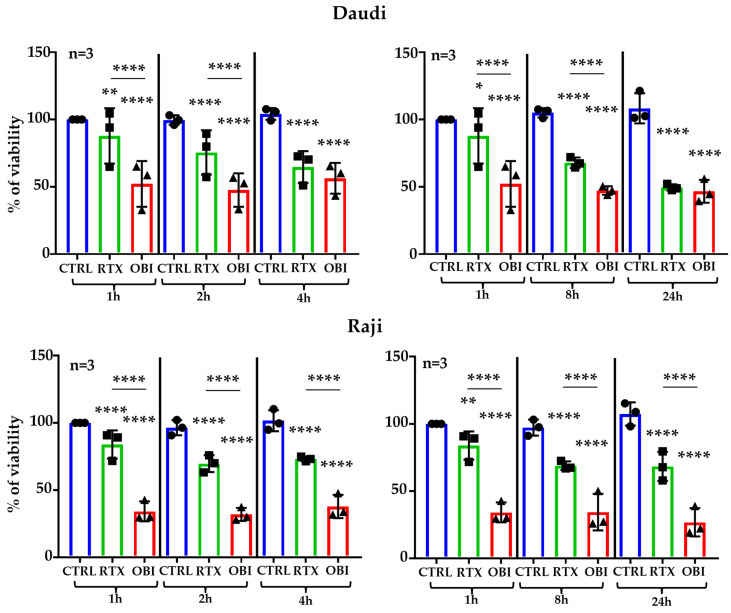
RTX and OBI induce quick direct cell death measured by Muse^®^ Cell Analyzer. The depicted cell lines (1 million/mL) were incubated with RTX or OBI (10 µg/mL) for the indicated times. Next, cells were incubated with the Muse^TM^ Count & Viability Assay kit and cell viability was directly analyzed in a Muse^®^ Cell Analyzer. The graphs display the percentage of viability after normalizing the viability at time 0 to 100%. The graphs display the mean ± SD of 3 independent experiments with biological replicates; * *p* < 0.05, ** *p* < 0.01, **** *p* < 0.0001; Tukey’s multiple comparisons test compared to control or as depicted in the graphics.

**Figure 2 cancers-15-01109-f002:**
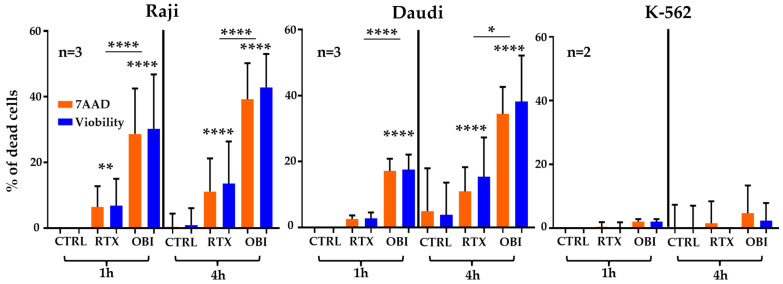
RTX and OBI induce quick direct cell death measured by 7AAD and Viobility^TM^ and analyzed by FACs. The depicted cell lines (1 million/mL) were incubated with RTX or OBI (10 µg/mL) for the indicated times. Then, cells were incubated with 7-AAD or Viobility^TM^ 405/452, following the provider’s instructions. Cells were washed and cell viability analyzed by FACs, using a Beckman Coulter Gallios cytometer. The graphs display the percentage of dead cells after normalizing the death at time 0 to 0%. Graphs show mean +/− SD of 3 (Raji and Daudi) or 2 (K-562) experiments performed in duplicate; * *p* < 0.05, ** *p* < 0.01, **** *p* < 0.0001; Tukey’s multiple comparisons test compared to control or as depicted in the graphics comparing groups to each other when using the same label.

**Figure 3 cancers-15-01109-f003:**
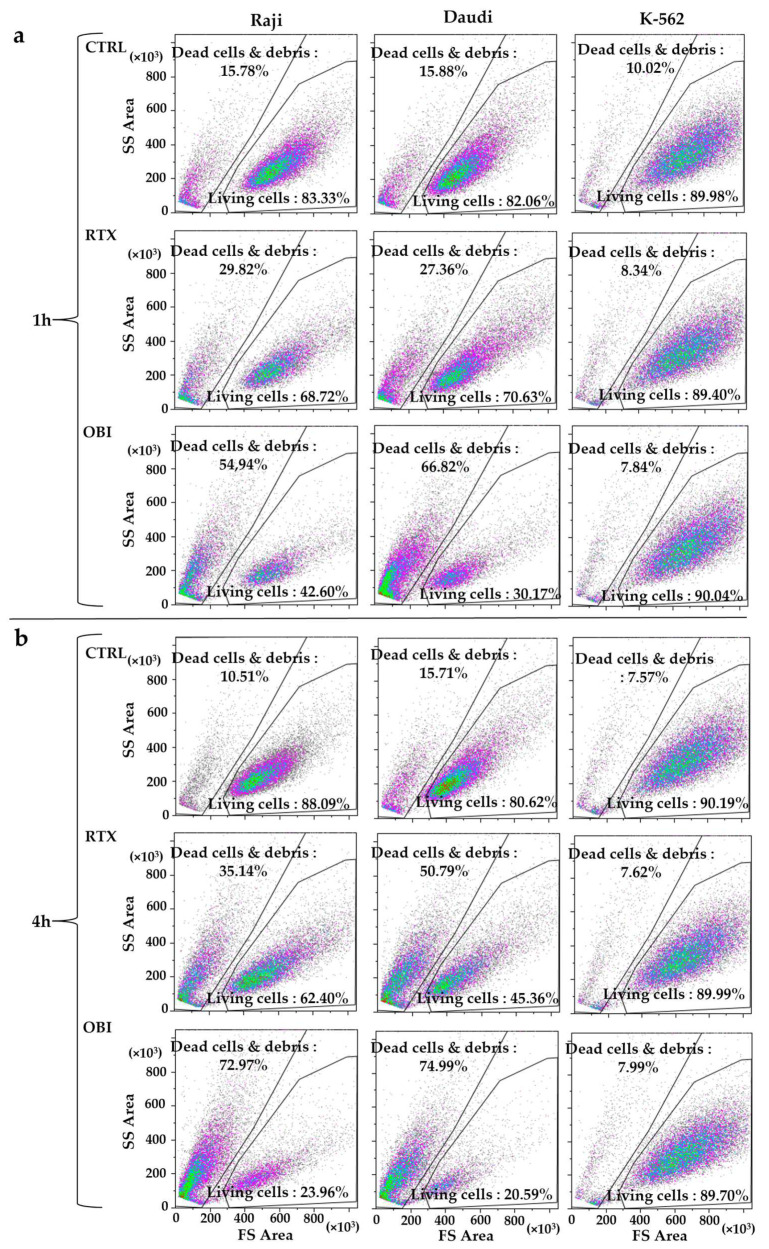
RTX and OBI induce a quick cell death measured by changes in cell morphology. The depicted cell lines (1 million/mL) were incubated with RTX or OBI (10 µg/mL) for the indicated times. Then, cells were directly analyzed by FACs with neither washing nor centrifugation steps. Doublets and very small debris were removed as described in Appendix A. (**a**) One hour post-treatment; (**b**) four hours post-treatment. Each graph displays 20,000 events.

**Figure 4 cancers-15-01109-f004:**
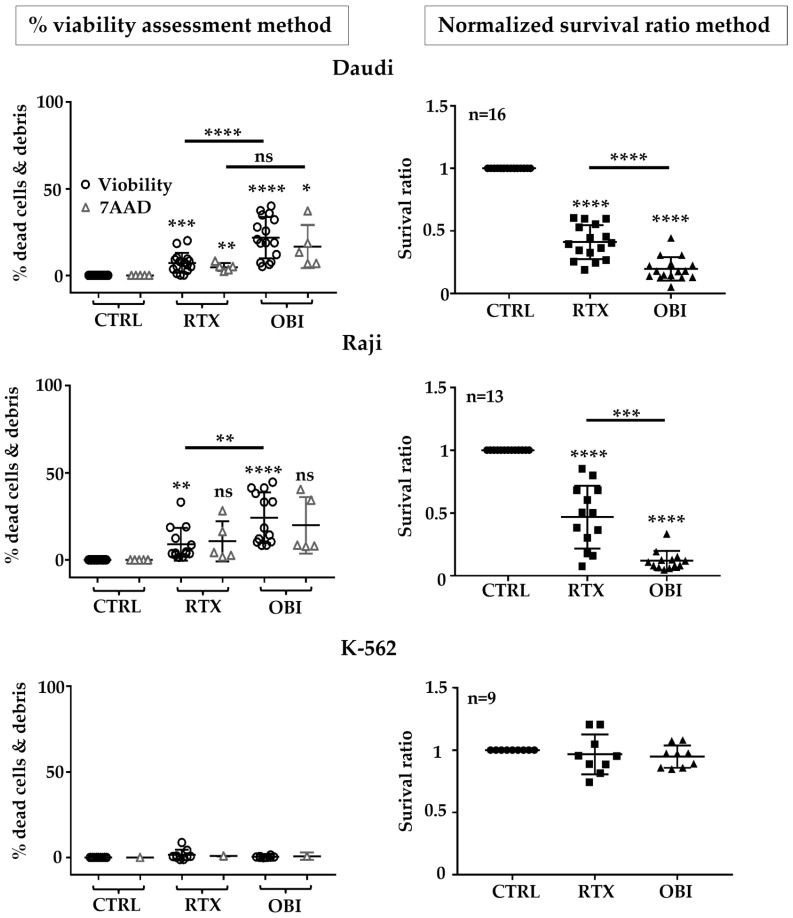
Using CD20 mAb-resistant MOLM-13 cells unveiled a high killing of target cells by RTX and OBI eight hours after treatment. Daudi, Raji, or K-562 cells were labeled with CTFR, whereas resistant MOLM-13 cells were with CFSE, following the provider’s instructions. Both cells were mixed at a 1-to-1 ratio and treated with RTX or OBI (10 µg/mL) before analyzing the number of cells in the mixed and the viability of the remaining cells by 7-AAD or Viobility^TM^ 405/452 staining (left graphs). The normalized survival ratio (right graphs) was calculated as depicted in Appendix A. Graphs show mean +/− SD of at least 5 experiments performed in duplicate; * *p* < 0.05, ** *p* < 0.01, *** *p* < 0.001, **** *p* < 0.0001; Tukey’s multiple comparisons test compared to control or as depicted in the graphics comparing groups to each other when using the same label.

**Figure 5 cancers-15-01109-f005:**
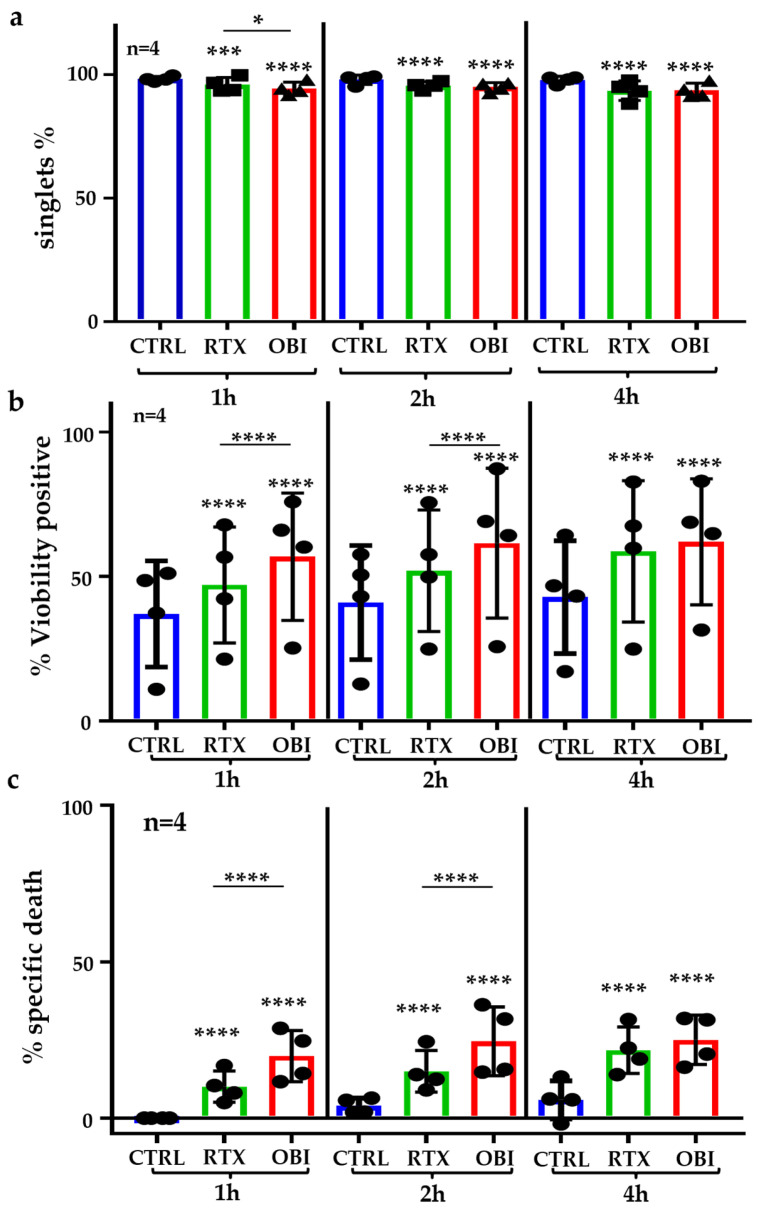
RTX and OBI induced cell aggregation and direct cell death on PBMCs from B-CLL patients. B-CLL PBMCs (1 million/mL) were thawed and directly incubated with RTX or OBI (10 µg/mL) for the indicated times. Next, cells were directly analyzed by FACs with neither washing nor centrifugation steps. (**a**) The percentage of singlets were quantified by FACs as described in Appendix A. After treatment, cells were washed and stained with an CD19 mAb to select B cells and Viobility^TM^ 405/452 to stain dead events. Cells were washed and analyzed by FACs using Beckman Coulter Gallios 3 Lasers. The graphs display the raw percentage of viability (**b**) or the treatment-specific cell death, which was obtained after subtracting the percentage of cell death from the control 1 h after incubation (**c**). Each graph represents mean +/− SD of 4 patients with measurements performed in duplicate. * *p* < 0.05, *** *p* < 0.001, **** *p* < 0.0001; Tukey’s multiple comparisons test compared to control or as depicted in the graphic.

**Figure 6 cancers-15-01109-f006:**
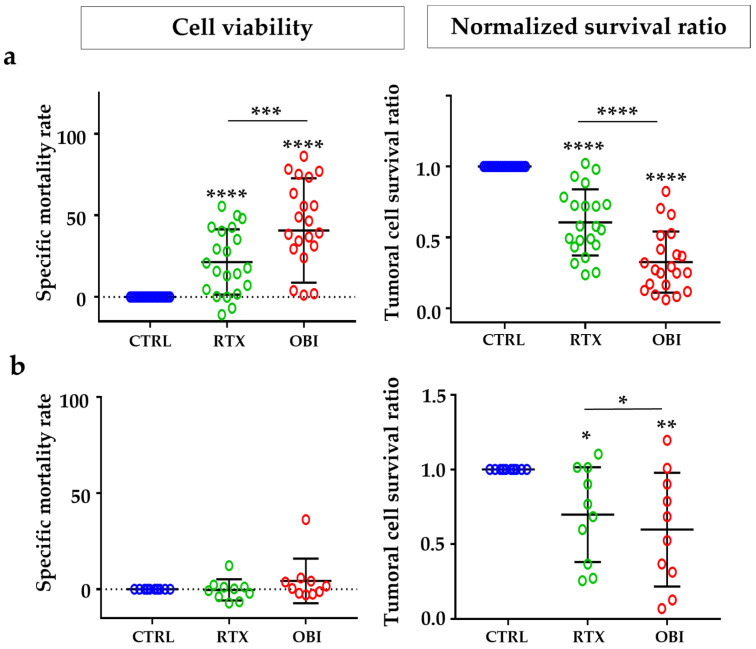
CD20 mAbs induced direct cell death on tumor cells from NHL and B-CLL patients. PBMCs from NHL (**a**) or B-CLL (**b**) patients at diagnosis were treated for 8 h with the depicted mAbs and tumor B cell death was examined by the method of cell viability (left) or normalized survival ratio (right) described in Appendix A. Each point represents one patient at diagnosis and performed in duplicate. Graphs show mean +/− SD, of 21 and 10 patients for NHL or B-CLL, respectively; * *p* < 0.05, ** *p* < 0.01, *** *p* < 0.001, **** *p* < 0.0001; Tukey’s multiple comparisons test compared to control or as depicted in the graphics.

**Figure 7 cancers-15-01109-f007:**
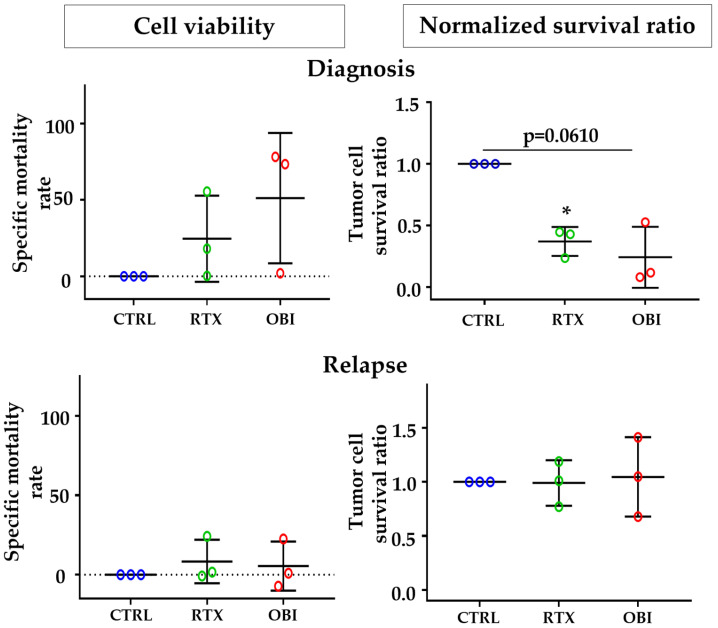
Tumor B cells from NHL and B-CLL patients at relapse are resistant to CD20 mAb-induced direct cell death. PBMCs from patients at diagnosis (top graphs) and after relapse (bottom graphs) were treated for 8 h with the depicted mAbs, and tumor B cell death was examined by the method of cell viability (left) or normalized survival ratio (right) described in Appendix A. Each point represents results for one patient at one time point (diagnosis or relapse) and was performed in duplicate. Graphs show mean +/− SD of 3 experiments; * *p* < 0.05; Tukey’s multiple comparisons test compared to control or as depicted in the graphics.

**Figure 8 cancers-15-01109-f008:**
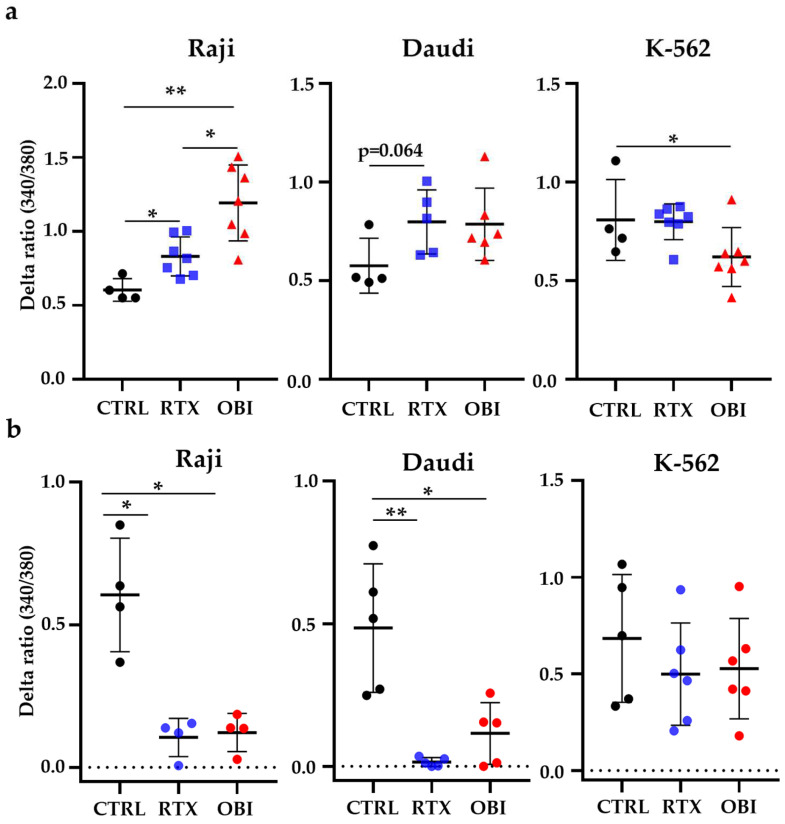
CD20 mAbs show antagonist effects on CCE depending on treatment duration. (**a**) Cells were treated with 10 µg/mL of RTX or OBI for 45 min and CCE was measured after addition of 5 mM of calcium. (**b**) RTX or OBI was applied just before adding 5 mM of extracellular calcium. Intracellular Ca2+ variations compared to the control condition were evaluated using the fluorescence emission of the Fura2 probe measured at 510 nm with an excitation light at 340 and 380 nm. Each graphic represents mean +/− SD of the number of experiments depicted in the graphs. Mann–Whitney was used to compare treatments. * *p* < 0.05; ** *p*< 0.01.

## Data Availability

Raw Data are available on demand.

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
