# Peer review of "Direct Cell Death Induced by CD20 Monoclonal Antibodies on B Cell Lymphoma Cells Revealed by New Protocols of Analysis"

_cancers, 2023, doi:10.3390/cancers15041109_

Round 1

Reviewer 1 Report

The manuscript named: Direct cell death induced by anti-CD20 mAbs on B cell lym-2 phoma cells revealed by new protocols of analysis 3, presents very comperhensive and inovative study with strong impact on general procedures in future in vitro studies on B cell lym-2 phoma cells or other cancer cell lines.

My opinion is that the manuscript is ready for publication in Cancers journal.

Author Response

We would like to thank the reviewer for her/his comments. The article has now been edited to eliminate English mistakes

Reviewer 2 Report

The authors examined killing of B-cells by CD20 monoclonal antibodies using different techniques. While the experiments seem to be valid and the results are interesting, the manuscript has a multitude of errors, inconsistencies and deficits.

Specific Points of Criticism and Suggestion for Alterations:

(1)  The English requires some thorough editing. Here I list only a few of the multitude of errors and mistakes (with no claim to completeness):

Line 27:  „than“ (instead of „that“)

Line 38:  „prior to increasing“ (instead of „previously to increase intracellular calcium levels“)

Line 87:  „peripheral“ (instead of „peripherical“)

Lines 234 and 243:  use „and“ in a text (instead of „&“).

Figure 4, Header and elsewhere:  „assessment“ (instead of „assesment“)

Line 345:  „complicate“ or „render it more complex“ (instead of „complexify“)

Figure 6, Legend:  „of 21 and 10 patients“ (instead of „21 or 10“)

Line 365:   „samples were collected from 3 NHL patients at their relapse after treatment“

Line 366:  „as“ (instead of „than“)

Line 367:  „at diagnosis“ (instead of „diagnostic“)

Line 369:  „while using“ (instead of „by using“)

Line 370 and elsewhere:  „extent“ (instead of „extend“)

Line 398:  delete „in acute“ (makes no sense and is not necessary)

Line 412:  „The use of“ (instead of „The used“)

Line 413:  „Whether our results“ (instead of „If“)

Line 425:  „technique“ (instead of „technic“)

Line 438:  „surviving cells“ (instead of „survival cells“)

Table S1:  „normalized“ (instead of „normalised“)

Figure S8, Legend:  „assess“ (instead of „asses“), „used as“ (instead of „use as“), „measured“ (instead of „mesured“)

Figure S9, Legend:  „which relies on the percentage“ (instead of „relies in looking to the %“)

Figures S10 and S11, Legends:  „Extent“ („extend“), „does not“ („do not“), „assessment“ („assesment“), „represents“ („represent“), „replicates“ („replicate“).

(2)  Title:  no abbreviations („mAbs“) in the title.

(3)  Lines 55 and 59:  What is the difference between Type I and Type II CD20 Abs? Maybe one explanatory sentence could help.

(4)  Line 65 and everywhere else:  This study uses both terms:  "CD20 antibody" and "anti-CD20 antibody". 

According to the Human Leukocyte Differentiation Antigen (HLDA) organization (www.hcdm.org) the following use is recommended, for example regarding CD2:  "CD2" is generally used to designate the molecule, and "CD2 antibody" is used to designate the antibody. 

So, not "anti-CD20 antibody", but „CD20 antibody“ will be the correct choice.

(5)  Line 88: At its first mention PBMCS must be defined here (presumably peripheral blood mononuclear cells); better use PBMC (without the "s").

(6)  Line 94:  Was the authenticity of cell lines checked and how?

(7)  Line 95:  Cell line MOLM-13 does not appear in the ATCC catalogue. Hence what was the source?

(8)  The chapters in Materials and Methods are numbered 1., 2., 3. ... – whereas the chapters in Results are numbered 3.1, 3.2, 3.3. ...

(9)  Line 151:  ("define this medium)"?

(10)  Chapter 11, Materials and Methods:  The authors switch back and forth from past tense to present tense:  „cells were seeded“ (line 172) – „cells are washed“ (line 175) – „entry is measured“ (line 179).

(11)  Lines 177 and 181:  „PBS“ in English (instead of „PSS“).

(12)  Statistical methods:  In Materials and Methods a chapter is missing on which statistical methods and tools have been used.

(13)  Everywhere in the manuscript:  Both K562 and K-562, MOLM13 and MOLM-13, BCLL and B-CLL are used. Some consistency please.

(14)  Figures and Supplementary Figures:  In the main text the abbreviation "OBZ" is used. Within the figures/supplementary figures and in the legends to the supplementary figures (but not in the legends of the regular figures) appears "OBI". I guess it would be easier to change the"OBZ" in the text (by search and replace) than to change the" OBI" in all the figures/supplementary figures.

(15)  In Chapter 3.4 no graphs are cited. The whole chapter is written like it would belong rather to "Materials and Methods" than to section "Results".

(16)  Line 371:  „CD20 antigen expression“.

(17)  Chapter 3.8:  The use of ionomycin, PMA and EGTA is not described in Materials and Methods.

(18)  Conclusion:  The Conclusion is thin and meager, more like a last sentence such as a „take-home message“. The Conclusion could be fleshed out more.

(19)  Supplementary Figures and Tables should be in the order in which they come up chronologically in the main text and not completely disordered as it is now, for example S4 appears first in line 123 - S2 in line 129 - S1 in line 196 - S3 in line 235, ...

(20)  Legend to Figure S4:  Periods and commas are mixed up, some sentences are incomplete.

Author Response

We  would like to thank the reviewer for her/his comments. The specific answers have been included in the enclosed document.
